# Examining Consumers’ Adoption of Wearable Healthcare Technology: The Role of Health Attributes

**DOI:** 10.3390/ijerph16132257

**Published:** 2019-06-26

**Authors:** Man Lai Cheung, Ka Yin Chau, Michael Huen Sum Lam, Gary Tse, Ka Yan Ho, Stuart W. Flint, David R Broom, Ejoe Kar Ho Tso, Ka Yiu Lee

**Affiliations:** 1Division of Business and Management, BNU-HKBU United International College, Zhu Hai 519000, China; 2Faculty of International Tourism and Management, City University of Macau, Macau, China; 3Faculty of Health and Wellbeing, Sheffield Hallam University, Sheffield S10 2BP, UK; 4Li Ka Shing Institute of Health Sciences, Chinese University of Hong Kong, Hong Kong, China; 5School of Nursing, University of Hong Kong, Hong Kong, China; 6School of Sport, Leeds Beckett University, Leeds LS6 3QS, UK; 7Academy of Sport and Physical Activity, Sheffield Hallam University, Sheffield S10 2BP, UK; 8Borneo Business School, North Borneo University College, Sabah 88400, Malaysian

**Keywords:** wearable healthcare technology, adoption intention, health belief, health information accuracy, privacy, consumer innovativeness, perceived usefulness

## Abstract

With the advancement of information technology, wearable healthcare technology has emerged as one of the promising technologies to improve the wellbeing of individuals. However, the adoption of wearable healthcare technology has lagged when compared to other well-established durable technology products, such as smartphones and tablets, because of the inadequate knowledge of the antecedents of adoption intention. The aim of this paper is to address an identified gap in the literature by empirically testing a theoretical model for examining the impact of consumers’ health beliefs, health information accuracy, and the privacy protection of wearable healthcare technology on perceived usefulness. Importantly, this study also examines the influences of perceived usefulness, consumer innovativeness, and reference group influence on the adoption intention of wearable healthcare technology. The model seeks to enhance understanding of the influential factors in adopting wearable healthcare technology. Finally, suggestions for future research for the empirical investigation of the model are provided.

## 1. Introduction

Wearable healthcare technology refers to smart electronic products that are incorporated into different types of accessories that can be attached to users’ bodies [1]. Examples include the Apple Watch, Fitbit, Samsung Gear and Mi Band wristband. These wearable technologies are regarded as one of the most promising areas in the Internet of Things (IoT) [2]. Wearable healthcare technology provides substantial impact on the wellbeing of citizens in our society, helping users to continuously monitor physiological parameters and health outcomes [3], which is useful in tracking and transforming users’ health information [4]. With the help of wearable healthcare technology, software is used to exchange data with peers, achieving a state of connected-self [5].

Wearable healthcare technology devices have a number of potential benefits for sports and healthcare industries [6]. For instance, using wearable healthcare technology devices can improve the accuracy of health information, fostering healthier behavior in individuals, significantly improving their health conditions, which in turn reduces healthcare costs [5]. When applied to a sports context, athletes can improve their sports performance by checking physiological data, such as their heart rate, running pace, and core temperature, along with other kinematic parameters such as joint angles or temporal parameters [7,8]. Sports players in various leagues, such as Euro-league Basketball, National Basketball Association (NBA), and National Football League (NFL), have adopted wearable healthcare technology devices to collect real-time data and provide a comprehensive overview of athletes [9].

Despite the promising developments of wearable healthcare technology, their adoption has lagged when compared to other well-established durable technology products, such as smartphones and tablets [6]. This is because of inadequate knowledge in the adoption intention of users of wearable healthcare technology [6,10,11,12] Wearable healthcare technology is still in the early stage of commercialization, with the majority of prior studies focusing on its technical development, resulting in inadequate understanding of its diffusion process [3,13]. Consumer adoption intention for smart technologies makes up a considerable part of marketing research, which provides meaningful implications for firms to speed up the diffusion process [14].

Notwithstanding, research on consumers’ adoption intention for wearable healthcare technology is limited, resulting in research gaps [5,15]. The majority of empirical studies employ the technology acceptance model (TAM), focusing on the technological antecedents of users’ adoption intention for wearable healthcare technology, leaving gaps in knowledge in a number of areas. First, prior studies have largely overlooked consumers’ perceived risks resulting from using wearable healthcare technology, including health and privacy risks [2], which in turn has resulted in unfavorable effects on users’ continuous usage intention [16]. Second, although wearable healthcare technology integrates health and technology attributes together to create value for consumers, limited studies, except for Chau et al. [4] and Zhang et al. [6], pay attention to consumers’ concerns in health, omitting the influence of consumers’ health beliefs in their adoption intention for wearable healthcare technology [4,6]. Third, prior studies have overlooked the influences of reference groups, such as the referrals from peers, family members, and brand community users, resulting in insufficient knowledge in promoting the adoption of wearable healthcare technology [4,12,17] Fourth, to the best of the authors knowledge, very few studies have focused on the synergistic effects of attributes related to health, technology, and consumers, along with reference group influence, as antecedents for the adoption of wearable healthcare technology. 

When consumers have strong and positive intentions to adopt wearable healthcare technology, its potential benefits will be more fully realized [4,10,18]. Thus, further research in examining the entire process of consumers’ adoption in wearable healthcare technology is necessary [19]. Seeking to address the aforementioned research gaps, the aim of this study is to examine the synergistic effects of health concerns, privacy concerns, consumer innovativeness, and influences from reference groups on consumers’ adoption intention for wearable healthcare technology. This study also contributes to sports, recreation, and healthcare industries by offering new ways to promote wearable healthcare technology, which in turn motivate individuals to be more engaged in sports, recreation, and therapy [12,20,21,22], as well as health literacy [23,24].

The remainder of this paper is organized as follows: First, we review the relevant literature to identify research gaps. Second, we present a theoretical model integrating consumers’ health concerns, privacy concerns, and consumer innovativeness, along with influences from reference group as antecedents for consumers’ adoption intention for wearable healthcare technology. Third, we present the research methodology to be used in testing the hypotheses derived from the theoretical model and analyze the data. Finally, we discuss the theoretical and managerial implications, incorporating limitations and directions for future research. 

## 2. Literature Review and Hypotheses Development 

### 2.1. Technology Acceptance Model (TAM)

The technology acceptance model (TAM) was introduced by Davis (1989) to examine why individuals accept a new technology and is regarded as one of the most important theories within the context of information system adoption [25,26,27,28] Empirical research adopts the TAM to examine consumers’ adoption intention in various contexts, such as online banking [29], mobile commerce [30], healthcare [31,32] sports websites [33] social networking sites [34], sports wearable technology [35] and wearable healthcare technology [10,17] The TAM emphasizes the importance of the perceived ease of use and the perceived usefulness in explaining consumers’ adoption intention for technological products [3,13,36,37,38]. The TAM is regarded as one of the most preferable theoretical models in explaining consumers’ intention to adopt new technology products [16,35]. Thus, we employ the TAM as the base model, along with additional constructs, in order to enhance the explanations and predictions of consumers’ acceptance behavior in the context of wearable healthcare technology [35]. In accordance to Zhang et al., we focus on the perceived usefulness to predict adoption intention, because perceived usefulness is particularly relevant to technologies that fit consumers’ lifestyle [4,17]. To extend the TAM, we simultaneously examine the impact of enabling factors adopted from the literature on consumers’ adoption intention for wearable healthcare technology, which will be discussed in the following sections.

### 2.2. Health Belief Model [HBM]

The health belief model [HBM] was initially adopted to predict individuals’ behavioral responses with acute or chronic diseases to the treatments received [39,40,41], and has been applied to guide general health behavior since then [42]. The HBM emphasizes the importance of the individual’s perceived susceptibility, severity, benefits, and barriers in predicting their health-related behavior [4]. Given the importance of information technology in explaining individual’s health-related behavior, the HBM was incorporated with the TAM to predict individual’s beliefs in using information technology for health-related purposes [39]. Indeed, individuals who believe that their health is suffering are motivated to use information technology to improve their health [39,42]. In the context of wearable healthcare technology, individuals who realize that their deviant health behaviors cause harm to their health are motivated to adopt wearable healthcare technology to develop their health management activities [6]. As such, in accordance to Zhang et al. [4], we incorporated the HBM with the TAM in this study to derive relatively new constructs, namely, the perceived health belief and health information concern, being posited as enabling factors in predicting consumers’ adoption intention for wearable healthcare technology, respectively. 

## 3. Theoretical Framework and Hypotheses Development 

### 3.1. Perceived Usefulness

Originating from the TAM, perceived usefulness describes a user’s subjective beliefs that using a specific information technology system would enhance his or her performance [37]. Perceived usefulness has been acknowledged as one of the most important drivers in predicting and explaining users’ intention in accepting information technology [5,14,16]. In particular, when users believe that the information technology devices are beneficial to their lives, such expected positive outcomes motivate their intention to adopt the information technology devices [35,43]. 

The importance of perceived usefulness in information technology acceptance has been confirmed in various contexts, such as in internet banking [44], smartphones [45], virtual reality [46], mobile exergames [47,48], mobile applications [49], and wearable technology [17]. Applied in the context of wearable healthcare technology, when devices such as healthcare applications, smart watches and sports-wearable technology products are perceived to be useful in improving consumers’ health status, such a positive expectation enhances consumers’ adoption intention [3,14,50]. We hypothesize that perceived usefulness is positively associated with the intention of consumers to adopt wearable healthcare technology, justifying our first hypothesis: Usefulness has a positive impact on consumers’ adoption intention for wearable healthcare technology. 

### 3.2. Health Belief

Originating from the HBM, health belief describes the consumers’ personal belief in the effectiveness of particular behaviors in improving their health status [4], being inextricably linked with consumers’ perceived usefulness and the adoption intention for wearable healthcare technology devices [4,6,39]. With the growing importance of healthcare, developers are creating smart technology products for the healthcare sector [16,51,52]. Indeed, one of the most important functions of wearable healthcare technology is to change consumers’ behavior in healthcare, which in turn improves their health status [4]. Particularly, wearable healthcare technology helps consumers to change their health behavior by providing data related to their health status, which is useful for consumers in fitness activity tracking and evaluating their performance in exercises, as well as planning customized exercises [44]. The aforementioned benefits may help consumers to improve their health status, being active and improving their quality of life [35]. 

Based on the promising functions of wearable healthcare technology, empirical studies use health belief as an additional construct to explain consumers’ acceptance intention for wearable healthcare technology [6]. Arguably, when a consumer has a stronger health belief, being involved in seeking ways to improve their health status, they may obtain information about the usefulness of wearable healthcare technology, which in turn strengthens the perceived usefulness of wearable healthcare technology in their minds [4]. Thus, we secondly hypothesize: Health belief has a positive impact on consumers’ perceived usefulness for wearable healthcare technology.

### 3.3. Health Information Accuracy

Health information accuracy refers to the degree in which consumers believe the information related to their health status provided by wearable healthcare technology is reliable and credible, being inextricably linked with the perceived usefulness of information technology products [2]. Arguably, when health information obtained from wearable healthcare technology is perceived to be accurate, consumers are willing to evaluate their health status and effectiveness of sports exercises by using healthcare wearable devices [4,53,54]. Indeed, the accuracy of health information provided by wearable healthcare technology has a positive impact on consumers’ willingness to make health-related decisions based on the information obtained from wearable healthcare technology devices [55], which in turn formulates consumers’ perceived usefulness [56]. As such, we arrive at our third hypothesis: Health information accuracy has a positive impact on consumers’ perceived usefulness for wearable healthcare technology.

### 3.4. Privacy Protection

Privacy protection refers to the degree to which consumers believe their personal information would not be misused or shared with others without their consent [2]. This is regarded as an important consideration for information technology adoption [10]. When users adopt wearable healthcare technology, their personal data relating to their health status is collected and saved on a database, inevitably raising consumers’ privacy concerns [12]. Empirical studies demonstrate the negative impact of consumers’ privacy concerns on their intention to accept information technology products, e.g., [2,10,57], justifying the importance of protecting consumers’ private data from unauthorized outflows [4]. As such, recent studies emphasize the importance of privacy protection in building a positive consumer attitude towards wearable healthcare technology products, e.g., [4,6,10]. In other words, privacy protection formulates users’ perceived usefulness [56], justifying our forth hypothesis: Privacy protection has a positive impact on consumers’ perceived usefulness for wearable healthcare technology.

Although perceived usefulness has been a rigorously tested construct for predicting consumers’ adoption intention for information technology products, it has been suggested that perceived usefulness should be incorporated with additional constructs in order to strengthen its predictive power for consumers’ adoption behavior for information technology [5,35]. In order to produce a more comprehensive investigation for consumer adoption behavior for wearable healthcare technology, two additional constructs, namely consumer innovativeness and reference group influence were added in the theoretical model to examine the antecedents of information technology adoption behavior. 

### 3.5. Reference Group Influence

Reference group influence refers to the extent to which consumer decision making is influenced by the perceptions of a reference group, which includes any important persons that shape consumers’ perceptions toward the focal product, such as parents, peers, or opinion leaders [58,59,60]. The influence of the reference group on the consumer decision making process is widely acknowledged in the empirical literature [61,62,63]. Prior to decision making, consumers may seek information and recommendation from reference groups with trustworthiness, or they may simply observe the behavior of individuals in the reference groups [64,65,66]. Similarly, in the context of information technology adoption, consumers tend to make their acceptance decisions towards information technology products based on the comments and recommendations from reference groups when the information technology products are relatively new to them [17]. 

Applied to the context of wearable healthcare technology, prior to decision making, consumers search for information and seek recommendation for the benefits of using wearable healthcare technology devices from reference groups via different channels, such as face-to-face communication, phone conversation, or social media platforms [4]. The adoption intention of consumers is shaped by reference group influence, because there are uncertainties from this kind of relatively new products [10]. Therefore, we arrive at our fifth hypothesis: Reference group influence has a positive impact on consumers’ adoption intention for wearable healthcare technology.

### 3.6. Consumer Innovativeness

Consumer innovativeness refers to the willingness of consumers to try out new information technology products, being inextricably linked with consumers’ general beliefs about information technology [35,67]. Individuals with better innovativeness appreciate the benefits of new technology, believe it is less troublesome, and have a higher propensity to embrace and use new technology products to accomplish their personal goals [35,68]. The higher the innovativeness of a consumer, the higher the propensity to recognize the benefits of new technology products [17].

Applying Roger’s [69] theory of the diffusion of innovations, empirical literature has confirmed that consumer innovativeness has a significant positive impact on consumers’ intention to accept information technology products [68,70]. In the context of wearable healthcare technology, empirical studies also posit that consumers with high innovativeness are able to handle uncertainty and have a greater adoption intention [4,12,17]. As such, it is logical to argue that consumer innovativeness is a significant predictor of adoption intention, especially for new technology products, like wearable healthcare technology devices. Thus, we arrive at our sixth hypothesis: Consumer innovativeness has a positive impact on consumers’ adoption intention for wearable healthcare technology. Figure 1 presents our research model.

## 4. Research Methodology

To test the research model in this study, we used a quantitative research method to conduct a survey in Hong Kong. We used measurement items adapted from previous studies with minor modifications in wording to fit our context. 

### 4.1. Development of Measurement Items

We used a self-administrated online survey (English language) to test the aforementioned hypotheses in the theoretical model to collect data from a convenience sample of consumers in Hong Kong using Qualtrics. We used measurement items adopted from previous studies to develop the survey questionnaire (see Table 1), where questions were measured on a 7-point Likert scale (1 = strongly disagree, 2 = disagree, 3 = somewhat disagree, 4 = neither agree or disagree, 5 = somewhat agree, 6 = agree, 7 = strongly agree). 

### 4.2. Data Collection

We collected responses from customers of wearable healthcare technology in Hong Kong. We promoted the survey using a convenience sampling approach, sending invitations to respondents via e-mail, Facebook, WeChat, and Instagram in Hong Kong. The data collection took place from October 1st, 2018, to January 25th, 2019, a total of 14 weeks. To ensure the validity of the survey, we assessed respondents’ knowledge of wearable healthcare technology by using several screening questions, such as “Have you ever used a wearable healthcare technology product?”, “Have you ever used a wearable technology product?”, and “Have you ever read product reviews about wearable healthcare technology products on website/social media platforms/blogs/forums?” Respondents without experience of wearable healthcare technology were excluded from the current study. After exclusion, we invited 310 consumers to participate in the survey. In total, 237 completed the survey. Sixty-six participants were discarded due to incomplete responses, resulting in a final sample of 171 participants, equivalent to a 55.2% response rate. Respondents were over 18 years old and users of wearable healthcare technology products. The sample was comprised of 55% male (45% female) respondents, with ages ranging from 18 to 65 years (mean = 26–30 years). A large proportion of respondents were aged between 22 and 26 (36.3%). Most respondents were university educated (50.9%), with experience in using wearable healthcare technology products, and a large proportion owned more than three wearable healthcare technology products (49.7%). 

## 5. Data Analysis and Results

In this study, we used partial least squares-structural equation modeling (PLS-SEM) for data analysis, using SmartPLS v3.28 (SmartPLS GmbH, Domainfactory GmbH, Ismaning, Germany) with the 5000-bootstrap procedure to assess (1) the measurement (outer) model and (2) structural (inner) model. We used PLS-SEM to do the data analysis after considering its unique advantages. PLS-SEM is considered as an appropriate technique for theory testing and theory confirmation [71,72,73], which fits the purpose of this study. Additionally, PLS-SEM is advantageous when the goal is to further advance the arguments of theoretical models, as is the case in this study [73]. Furthermore, PLS-SEM is appropriate for studies with smaller sample sizes, such as in this study [73]. 

### 5.1. Measurement Model

Following Hair et al. [73], we tested the reliability of the measurement items by checking their factor loadings, Cronbach’s alpha, and composite reliability (CR) in the measurement model. As reported in Table 1, the loadings of all items were greater than 0.813 and significant (i.e., the factor loadings of items were ranged between 0.813 to 0.953, with *p* ≤ 0.000), whereas Cronbach’s alpha and the CR of all of the constructs were greater than 0.845, which is well above the recommended 0.70 threshold [73], confirming the reliability of the measurement items and constructs. In addition, as reported in Table 2, we tested the convergent validity of the model using the average variance extracted (AVE), as all of the AVE values were larger than the recommended value of 0.5, confirming the convergent validity of the constructs. Lastly, the discriminant validity was tested by using the Fornell and Larcker criterion, where the AVE square roots were larger than the corresponding correlations, demonstrating discriminant validity [73].

### 5.2. Structural Model

We used SmartPLS v3.28 with the 5000-bootstrap procedure, to test the hypotheses in our theoretical model. Figure 2 presents the results of the structural model. 

The hypotheses were tested by examining the *t*-values, *p*-values, standardized coefficient beta values and the coefficient of determination (R^2^ value). A hypothesis was accepted when the *t*-value was larger than the critical value (i.e., *t* ≥ 1.96, *p* ≤ 0.05), being marginally accepted when the *t*-value was larger than the critical value (i.e., *t* ≥ 1.67, *p* ≤ 0.10), using a two-tailed test. 

As reported in Figure 2, the results support five of the six hypotheses. Regarding the antecedents of perceived usefulness, the impact of health information accuracy on perceived usefulness was the strongest (β = 0.516, *p* < 0.001), followed by health belief (β = 0.206, *p* < 0.05), supporting H2 and H3. However, the impact of privacy protection on perceived usefulness was weak and non-significant (β = 0.170, *p* > 0.05), rejecting H4. 

Regarding the antecedents of adoption intention, the impact of perceived usefulness on adoption intention was strongest and significant (β = 0.335, *p* < 0.001), followed by reference group influence (β = 0.331, *p* < 0.001) and consumer innovativeness (β = 0.293, *p* < 0.010). Therefore, H1, H5 and H6 are supported.

Furthermore, the results also presented the indirect effects of the exogenous variables, including health belief, health information accuracy and privacy protection on adoption intention. In particular, the indirect effects of health information accuracy (β = 0.173, *p* < 0.010) and health belief (β = 0.069, *p* < 0.05) on adoption intention were significant, whilst the indirect effects of privacy protection (β = 0.057, *p* > 0.10) on adoption intention was non-significant. Table 3 summarizes the results of the hypotheses testing of this study. 

In order to test the importance of the exogenous variables in the theoretical model, we used Cohen’s *f*^2^ analysis to test the effect size of the exogenous variables. According to Cohen [74], the *f*^2^ values were assessed as *f*^2^ 0.02 (i.e., *f*^2^ ≦ 0.02), 0.15 (i.e., *f*^2^ ≦ 0.15), and 0.35 (i.e., *f*^2^ ≦ 0.35), representing small, medium, and large effects of the exogenous latent variables, respectively [74]. The results show that the effect size of health information accuracy (*f*^2^ = 0.258) and reference group influence (*f*^2^ = 0.210) were medium to large, while the effect size of perceived usefulness (*f*^2^ = 0.140), consumer innovativeness (*f*^2^ = 0.111), and health belief (*f*^2^ = 0.075) were small to medium. The results reveal that the exogenous variables have considerable effects in the theoretical model [74]. 

Importantly, we evaluate the explanatory power of the research model by assessing the R^2^ values (see Figure 2). The R^2^ values for the perceived usefulness and adoption intention were 0.628 and 0.749, respectively. This highlights that these exogenous constructs explain 62.8% and 74.9% of the variation in perceived usefulness and adoption intention, respectively. In short, as the R^2^ values exceed the recommended criterion benchmark (i.e., > 0.10), the results suggest that the research model explains a meaningful amount of variation in the endogenous variables, in line with [73]. 

## 6. Discussion

### 6.1. Theoretical Implications

This study provides several theoretical implications and complements the existing literature in the area of wearable healthcare technology by identifying and empirically examining the synergistic effects of health related attributes, consumer attributes, and social influences on the behavioral intention of consumers towards wearable healthcare technology. Although wearable healthcare technology provides distinctive advantages in improving consumers’ healthcare efficiency, attributes related to health and privacy have been overlooked in prior studies [3,52,75]. This study complements the existing literature by examining the impacts of health belief, health information accuracy, and privacy protection on consumers’ perceived usefulness of wearable healthcare technology. The findings reveal that health belief and health information accuracy are significant antecedents for consumers’ perceived usefulness, and their indirect effect on consumers’ adoption intention is significant. Surprisingly, we found that the impact of privacy protection on perceived usefulness is non-significant, suggesting that privacy protection is not an antecedent of perceived usefulness for wearable healthcare technology. 

Second, the findings reveal that perceived usefulness, consumer innovativeness, and reference group influence are important antecedents that have a significant positive impact on consumers’ adoption intention for wearable healthcare technology. In particular, perceived usefulness had the strongest influence on adoption intention, greater than consumer innovativeness and reference group influence, confirming the findings of previous studies that adopted perceived usefulness to explain adoption intention in the context of information technology products [13,26,76,77]. The findings also reveal that consumer innovativeness and reference group influence are critical factors in strengthening consumers’ intention to adopt wearable healthcare technology. The results suggest that technology readiness and recommendations from the reference group greatly influence consumers’ intention to adopt wearable healthcare technology.

In line with recent studies related to wearable healthcare technology [4,6,78], this study proposes that consumers’ adoption intention requires not only perceived usefulness, but also variables related to healthcare, the reference group, and consumers’ perceptions. As such, compared with previous studies using the TAM to examine the adoption intention of wearable healthcare technology, e.g., [4,10], this study proposed the consumers’ process of such acceptance. In other words, this study reveals that health attributes, including health belief and health information accuracy, positively relate to consumers’ perceived usefulness, which in turn, contributes to consumers’ adoption intentions. 

### 6.2. Managerial Implications

This study provides several managerial implications for marketers and developers of wearable healthcare technology which may be used to better promote their products. First, the results of this study suggest that perceived usefulness has a strong and significant influence on consumers’ adoption intention for wearable healthcare technology. Therefore, marketers are recommended to persuade consumers by communicating the benefits of using wearable healthcare technology, such as through fitness checking, the evaluation of sports exercises, and checking health status [35]. 

Second, the findings of this study suggest that health information accuracy and health belief play critical roles in building consumers’ perceived usefulness. As such, developers should ensure the accuracy of health data to be provided by wearable healthcare technology, eliminating consumers’ concerns arising from fear of mismanaging their health because of inaccurate data obtained from wearable healthcare technology [2]. In addition, marketers should promote the usefulness of wearable healthcare technology in improving consumers’ health, whilst addressing health concerns [6]. Once consumers perceive that the data obtained from wearable healthcare technology devices are accurate, along with their effectiveness in improving health status, their perceived usefulness would be strengthened, in turn providing positive indirect effects for adoption intention. 

Third, the results of this study suggest that consumer innovativeness and reference group influence are significant predictors of adoption intention. Thus, marketers should communicate the innovative features of wearable healthcare technology, including the provision of messages about sleep quality, resting heart rate, steps taken, and workout intensities [35]. Importantly, marketers should encourage reference groups of various social media communities to publish information about the benefits of using wearable healthcare technology [4,62]. The above-mentioned recommendations are useful in strengthening adoption intentions for wearable healthcare technology. 

## 7. Limitation and Directions for Future Research

Despite the useful findings about consumers’ adoption behavior in this study, some limitations persist, suggesting directions for future research. First, this study has been conducted in Hong Kong, limiting its generalizability on a global scale. Thus, future research should replicate this study in other countries, providing comparisons between more countries in different regions with diverse cultures [17]. This could include, though is not restricted to, countries in North America, Latin America, and Africa. Second, although the constructs in the theoretical model provide significant explanatory power (i.e., R^2^ = 0.749) about consumers’ intention to adopt wearable healthcare technology, future research should incorporate additional variables in our theoretical model, such as perceived trust, hedonic motivation, and utilitarian motivation, along with the conceptualization of health belief as multi-dimensional construct, in order to understand consumers’ intention to accept wearable healthcare technology more comprehensively. Third, this study concentrated on perceived usefulness in explaining consumers’ adoption intention of wearable healthcare technology. Thus, future research is recommended to include perceived ease of use in the TAM to explain consumers’ adoption intentions. Finally hindering factors, such as health concerns, privacy concerns [2], and risk [12] should be incorporated in future research.

## 8. Conclusions

With promising developments in innovative technology products and applications, healthcare wearable technology devices are expected to provide new ways to cope with issues related to health [78]. For example, integrating healthcare wearable technology with mobile health applications that monitor sports exercises can encourage individuals to do more sports exercises, which is useful in decreasing the negative effects of their sedentary lifestyle, as well as maintaining their health and wellness [35,78]. As such, we empirically tested a research model to investigate the antecedents of adoption intention of healthcare wearable technology. Our findings revealed that health belief and health information accuracy had significant impact on perceived usefulness, which in turn driving adoption intention. We also asserted that reference group influence and consumer innovativeness are critical drivers of adoption intention. The findings of this paper provide meaningful implications for both academics and managers. 

## Figures and Tables

**Figure 1 ijerph-16-02257-f001:**
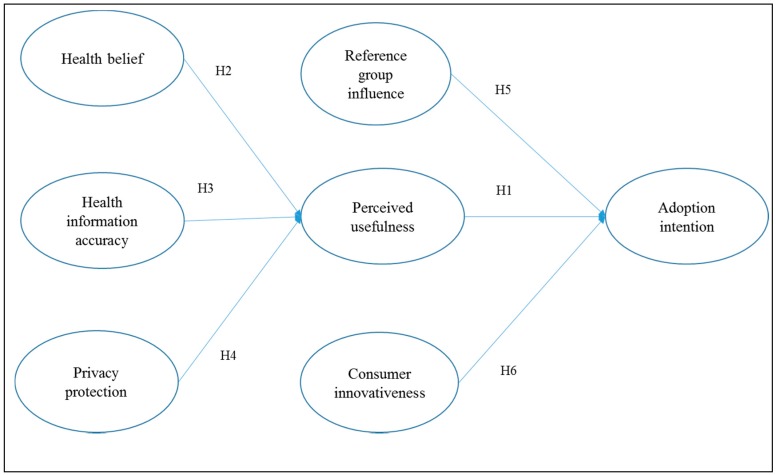
Research model.

**Figure 2 ijerph-16-02257-f002:**
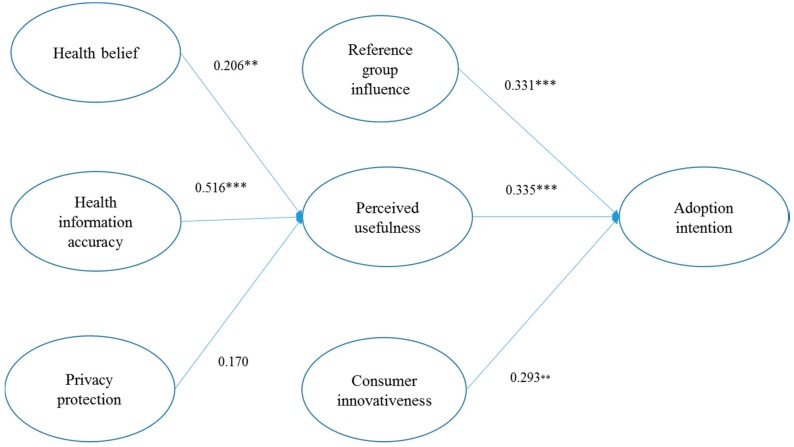
Results of the theoretical model. Paths were considered significant at ** *p* < 0.01, *** *p* < 0.001.

**Table 1 ijerph-16-02257-t001:** Outer model results.

Construct	Loading	*t*-Value	Alpha	Composite Reliability
Health Belief [4]			0.874	0.914
I realize that bad living habits will cause harm to my health	0.841	24.900		
I perceive that bad living habits will cause harm to my health	0.840	25.116		
I hope I can change my bad habits and thus to minimize damage to health	0.883	35.881		
I think I can improve my health status effectively in many ways like sports	0.844	26.528		
Health Information Accuracy [2,4]			0.873	0.940
The health information provided by the wearable healthcare technology is accurate	0.942	93.209		
The health information provided by the wearable healthcare technology is trustworthy	0.941	81.381		
Privacy Protection [2,4]			0.886	0.946
The wearable healthcare technology has provided adequate protection of my personal health information	0.948	115.195		
The supplier of healthcare wearable device will not share my personal health information with other entities without my authorization	0.946	116.074		
Perceived Usefulness [3]			0.925	0.952
Using the healthcare wearable device would be useful in my personal health management	0.928	91.174		
Using the healthcare wearable device would help me develop healthy habits	0.939	69.377		
Using the healthcare wearable device would help me maintain healthy status	0.931	74.235		
Consumer Innovativeness [67]			0.845	0.907
I like to experiment with new things and products	0.912	63.330		
I think a new way of life and a new pattern of consumption is a kind of progress compared with the past	0.897	49.085		
In general, I am among the first in my circle of friends to use a new technological product or service when they appear	0.813	26.787		
Reference Group Influence [4]			0.885	0.946
I often take notice of health information related to healthy habits and status released by my friends on Facebook/Instagram/WeChat.	0.950	134.522		
I often browse health information and health news shared by my friends on Facebook/Instagram/WeChat.	0.944	88.491		
Adoption Intention [3,4,12]			0.938	0.960
I am interested in using the healthcare wearable device	0.946	107.358		
I plan to adopt the healthcare wearable device in the future	0.942	51.599		
I will develop healthy habits with the healthcare wearable device in the future	0.942	84.637		

**Table 2 ijerph-16-02257-t002:** Construct correlation matrix and average variance extracted (AVE).

Constructs	AI	CI	HB	HIA	PU	PP	RGI	AVE	Square Root of AVE
Adoption Intention (AI)	1							0.890	0.943
Consumer Innovativeness (CI)	0.790	1						0.766	0.875
Health Belief (HB)	0.555	0.592	1					0.726	0.852
Health Information Accuracy (HIA)	0.792	0.709	0.577	1				0.897	0.947
Perceived Usefulness (PU)	0.802	0.808	0.596	0.768	1			0.870	0.933
Privacy Protection (PP)	0.691	0.623	0.543	0.783	0.685	1		0.908	0.953
Reference Group Influence (RGI)	0.764	0.683	0.503	0.717	0.696	0.574	1	0.897	0.947

**Table 3 ijerph-16-02257-t003:** Results of hypotheses testing.

Hypotheses	Results
H1: Perceived usefulness has a positive impact on consumers’ adoption intention of wearable healthcare technology.	Supported
H2: Health belief has a positive impact on consumers’ perceived usefulness of wearable healthcare technology.	Supported
H3: Health information accuracy has a positive impact on consumers’ perceived usefulness of wearable healthcare technology.	Supported
H4: Privacy protection has a positive impact on consumers’ perceived usefulness of wearable healthcare technology.	Rejected
H5: Reference group influence has a positive impact on consumers’ adoption intention of wearable healthcare technology.	Supported
H6: Consumer innovativeness influence has a positive impact on consumers’ adoption intention of wearable healthcare technology.	Supported

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
