# Peer review of "Examining Consumers’ Adoption of Wearable Healthcare Technology: The Role of Health Attributes"

_ijerph, 2019, doi:10.3390/ijerph16132257_

Round 1

Reviewer 1 Report

Relevance to journal

Overall the paper addressed an interesting topic that is of interest to the readership of the journal.

Abstract

The abstract provides a good summary of the paper.

Introduction

The authors describe a gap in the literature in our understanding of the reasons for the adoption of healthcare wearable technology and they carry out a study that looks at influential factors in adopting healthcare wearable technology.

Literature review

The authors provide a literature review covering a wide range of literature, that suggests that there already is a substantial body of research on varied aspects of healthcare wearable technology, but the authors believe it would be useful to integrate additional variables into the basic TAM model to provide  better understanding of health belief and other variables that are relevant to the adoption of these wearable technologies.

Following a review of the literature on the TAM model and Health beliefs models, the authors present a model that integrates these, along with consumers’ health concerns, privacy concerns, innovativeness and influences from reference group as antecedents of consumers’ adoption intention of healthcare wearable technology.

While it was clear why TAM and the Health belief model were being used and these were discussed a lot (along with other variables), the authors’ choice of specific items for their implementation of the TAM and the Health belief models could be explained more clearly as they do not seem to use the standard constructs. The authors could make it clearer why they concentrated on perceived usefulness and excluded perceived ease of use as one of the key TAM variables – maybe perceived usefulness is more relevant here or the literature suggests that perceived ease of use is just not as predictive? But the authors should make this clear. With respect to the health belief model, the authors do not seem to use the key variables associated with the health belief model and collapse ll items into one scale? The authors do mention subscakes on page 3 - perceived susceptibility, severity, benefits and barriers, but again they could make it clearer why they selected the items they did, as they did not seem to me to map onto the variables outlined in the health belief model - susceptibility, severity, benefits and barriers. The authors need to make it clearer how the items they used map onto the health belief model.

I would have thought that the authors might have considered the theory of planned behaviour relevant in looking at the adoption of wearable technology. TPB looks at attitudes (to wearable technology), subjective norms and perceived behavioural control. The authors look at influences from reference group which is similar to subjective norms, but it would have been useful to include an attitudes variable. The authors do return to discuss other relevant variables in the discussion, such as perceived ease of use, perceived trust, and hedonic motivation. They might have considered the relevance of TPB here including attitudes to wearable health technology.

Methods

I felt this could have been better organised. I would expect to see a summary statement of the study design at the start of the methods section.

There is a reasonable account of how participants were recruited, although normally the description of the participants would precede the account of the materials. I would expect to see more information about the measures used in the methods too, especially given the problems of mapping the theoretical constructs on the actual measures used in this study. It would be useful also to include a rationale for the small number of items used in each scale. A short procedure section would be useful too.

Results

I would have thought this should be a separate section.

The results were explained clearly. Validity of measures used and contribution of items seemed good.

Structural equation modelling is a highly appropriate model providing both the measurement model and the structural model.

Given the items used in the analyses, the model seemed a good fit.

It would have been informative to see some discussion of competing models, for example with privacy protection removed given its lack of significance. Also since the model was trying to bring 2 other models together (TAM and HBM) was there a better way of doing this?

In Data collection section on page 7, mean age 26-30 years is a wide ranging mean! Was it a category?

Referencing

The article is grounded in relevant literature and well referenced.

The authors need to check all references for use of capital letter for journals.

Some references missing e. g. Kim & Chiu, 2019).

Typographical errors

There are several typographical and expressional errors in the script that need to be corrected. Many of these are words run together.

P2 line 45- 2012As such,

P2 line 45- ofthe

Verbal expression

Too many “arguablys”! Eg line 61 “Arguably, the majority of empirical studies employ the technology acceptance model (TAM)”, We would expect the authors to know whether the TAM model is the most prevalent?

Author Response

Reviewer 1’s Comments

Response to Reviewer 1

Overall the paper addressed an interesting topic that is of interest   to the readership of the journal.

Thank you for   your positive comments.

Abstract

The abstract provides a good summary of the paper.

Thank you for   your positive comments.

Introduction

The authors describe a gap in the literature in our understanding of   the reasons for the adoption of healthcare wearable technology and they carry   out a study that looks at influential factors in adopting healthcare wearable   technology.

Thank you for   your positive comments.

Literature review

The authors provide a literature review covering a wide range of   literature, that suggests that there already is a substantial body of   research on varied aspects of healthcare wearable technology, but the authors   believe it would be useful to integrate additional variables into the basic   TAM model to provide better understanding of health belief and other   variables that are relevant to the adoption of these wearable technologies.

Following a review of the literature on the TAM model and Health beliefs   models, the authors present a model that integrates these, along with   consumers’ health concerns, privacy concerns, innovativeness and influences   from reference group as antecedents of consumers’ adoption intention of   healthcare wearable technology.

While it was clear why TAM and the Health belief model were being used   and these were discussed a lot (along with other variables), the authors’   choice of specific items for their implementation of the TAM and the Health   belief models could be explained more clearly as they do not seem to use the   standard constructs. The authors could make it clearer why they concentrated   on perceived usefulness and excluded perceived ease of use as one of the key   TAM variables – maybe perceived usefulness is more relevant here or the   literature suggests that perceived ease of use is just not as predictive? But   the authors should make this clear. With respect to the health belief model,   the authors do not seem to use the key variables associated with the health   belief model and collapse ll items into one scale? The authors do mention   subscakes on page 3 - perceived susceptibility, severity, benefits and   barriers, but again they could make it clearer why they selected the items   they did, as they did not seem to me to map onto the variables outlined in   the health belief model - susceptibility, severity, benefits and barriers.   The authors need to make it clearer how the items they used map onto the   health belief model.

I would have thought that the authors might have considered the theory   of planned behaviour relevant in looking at the adoption of wearable   technology. TPB looks at attitudes (to wearable technology), subjective norms   and perceived behavioural control. The authors look at influences from   reference group which is similar to subjective norms, but it would have been   useful to include an attitudes variable. The authors do return to discuss   other relevant variables in the discussion, such as perceived ease of use,   perceived trust, and hedonic motivation. They might have considered the   relevance of TPB here including attitudes to wearable health technology.

Thank you for   your comments.

We have   amended the manuscript according your comments. Please refer to the   amendments in blue.

We do not use   Perceived Ease of Use in the model because perceived usefulness is more   relevant to adoption intention of technological products that fit consumers’   lifestyle. Additionally, when Perceived Ease of Use is added in the model,   the majority of paths become non-significant.

We have   explained the use of constructs, please refer to the amendments in blue for   more details.

Additionally,   given the limitation in the scope of our study, we have added a paragraph to   discuss our limitation related to the concentration on perceived usefulness   and the conceptualization of health belief as uni-dimensional construct.

Methods

I felt this could have been better organized. I would expect to see a   summary statement of the study design at the start of the methods section.

There is a reasonable account of how participants were recruited,   although normally the description of the participants would precede the   account of the materials. I would expect to see more information about the   measures used in the methods too, especially given the problems of mapping   the theoretical constructs on the actual measures used in this study. It would   be useful also to include a rationale for the small number of items used in   each scale. A short procedure section would be useful too.

Thank you for   your comments. We have included a summary to describe the methodology of our   research. Please refer to the amendments.

Results

I would have thought this should be a separate section.

The results were explained clearly. Validity of measures used and   contribution of items seemed good.

Structural equation modelling is a highly appropriate model providing   both the measurement model and the structural model.

Given the items used in the analyses, the model seemed a good fit.

It would have been informative to see some discussion of competing   models, for example with privacy protection removed given its lack of   significance. Also since the model was trying to bring 2 other models   together (TAM and HBM) was there a better way of doing this?

In Data collection section on page 7, mean age 26-30 years is a wide   ranging mean! Was it a category?

Thank you for   your comments. We have described the impact of the constructs in the model.   Please refer to the result section.

Referencing

The article is grounded in relevant   literature and well referenced.

The authors need to check all references for   use of capital letter for journals.

Some references missing e. g. Kim & Chiu,   2019).

Typographical errors

There are several typographical and expressional   errors in the script that need to be corrected. Many of these are words run   together.

P2 line 45- 2012As such,

P2 line 45- ofthe

Verbal expression

Too many “arguablys”! Eg line 61 “Arguably,   the majority of empirical studies employ the technology acceptance model   (TAM)”, We would expect the authors to know whether the TAM model is the most   prevalent?

Thank you for   your comments. We have amended the references and typos.

Reviewer 2 Report

The paper presents a research model based on TAM and presents a survey to find the factors that support buyers' intention to buy health IoT products. There is a rich body of references that is used to support the authors' research. However, the literature review is at some points unstructured; e.g., when similar concepts are supported by different citations (e.g., lines 60-80 and 120-190). Citations mentioned, but not in the reference list: e.g., Kim&Chiu 2019. 

The authors use TAM based on attitudes that are not necessarily based on direct experiences. This can possibly have an impact on the beliefs, as for example, the construct of privacy protection may illustrate. Many customers have not yet experienced the disadvantages of privacy breaches in health care products (=lower awareness) while many of these products seem not to fulfil the expected expectations. For example, health care products have been banned from sale in some regions (e.g., some health watches for children) due to privacy issues. Low privacy awareness and assuming a belief based on advertisements for the products might not be in accordance with reality. Such impact needs to be discussed.

Further, the impact of privacy is not mentioned in the managerial implications (probably as a consequence of the low impact factor). However, not putting managerial emphasis on privacy may be a path to market failure (as the example with the health watches for children shows). From the results of the study, I agree that the authors cannot conclude with an impact to the managerial implications. Instead, this should trigger a discussion how your research model relates to the real world, and how to possibly mend this.

For privacy protection, it is not sufficiently argued for why this construct is in the compound of perceived usefulness in your research model (instead of direct impact on the adoption intention.).

The use of TAM has been disputed for a while. The authors should address the concerns of authors that published critiques of TAM, and argue that their research model does not suffer from the shortcomings mentioned there. See: Benbasat&Barki (2007): Quo vadis, TAM? J. Assoc. Inf. Systems; and for health care: Sniehotta, Presseau & Araujo-Soares (2014): Time to retire the theory of planned behaviour; Health Psychology Review.

When introducing the constructs, please make clear which constructs are compound constructs. The description of the constructs should be improved, and the differences between them should be pointed out. (e.g., 3.1 and 3.2 use partially similar wording)

The authors present six hypotheses; it is not explicitly outlined in the paper which of these hypotheses are adopted and which are rejected. Further, for each of these adoption/rejection decisions, the consequences and reasons should be discussed.

There are several paragraphs that are unclear or imprecise due to language issues (long and strange sentences).

* line 32: IoT (not: IOT)

* line 33: wellbeing of our society: probably you mean: wellbeing of citizens in our society.

* line 36: unclear; please explain why the connected-self is relevant here.

* line 38: sentence is unclear. You say: the devices have benefits for industries. Is this correct?

* line 42: remove word "elite" (also non-elite athletes improve their performance using IoT devices)

* Line 45: incomplete sentence; why only professionals?

* in several occasions, the word "arguably" is used. This does not always give meaning. For instance, lines 60, 72, 79. Btw., the entire sentence starting in line 60 is not comprehensible.

* line 90: a small paragraph presenting the contribution of the paper would be welcome.

* line 110: what is "overlooking of interpersonal influence" ?

* line 117: incomprehensible sentence

* line 126: incomprehensible sentence

* line 120: that their health is suffering: please rephrase

In Table 1: Please check consistency (this looks suspicious, but may be ok):

why are loading factors for both lines for H.I.Accuracy the same?

why are loading factors for both lines for Privacy Protection the same?

Author Response

Reviewer 2’s Comments

Response to Reviewer 2

The paper presents a research model based on   TAM and presents a survey to find the factors that support buyers' intention   to buy health IoT products. There is a rich body of references that is used   to support the authors' research. However, the literature review is at some   points unstructured; e.g., when similar concepts are supported by different   citations (e.g., lines 60-80 and 120-190). Citations mentioned, but not in   the reference list: e.g., Kim&Chiu 2019. 

Thank you for   your comments. We have amended the references.

The authors use TAM based on attitudes that   are not necessarily based on direct experiences. This can possibly have an   impact on the beliefs, as for example, the construct of privacy protection   may illustrate. Many customers have not yet experienced the disadvantages of   privacy breaches in health care products (=lower awareness) while many of   these products seem not to fulfil the expected expectations. For example,   health care products have been banned from sale in some regions (e.g., some   health watches for children) due to privacy issues. Low privacy awareness and   assuming a belief based on advertisements for the products might not be in   accordance with reality. Such impact needs to be discussed. 

Thank you for   your comments. The context of the study is Hong Kong, and we have explored   respondents’ experience in using healthcare wearable technology by screening   questions. We have tried our best to explore the issues you mentioned.

Further, the impact of privacy is not   mentioned in the managerial implications (probably as a consequence of the   low impact factor). However, not putting managerial emphasis on privacy may   be a path to market failure (as the example with the health watches for   children shows). From the results of the study, I agree that the authors   cannot conclude with an impact to the managerial implications. Instead, this   should trigger a discussion how your research model relates to the real   world, and how to possibly mend this. 

Thank you for   your comments. We have mentioned the omission of privacy issues in our   limitation.

For privacy protection, it is not   sufficiently argued for why this construct is in the compound of perceived   usefulness in your research model (instead of direct impact on the adoption   intention.).

Thank you for   your comments. We have discussed this issue in our limitation.

The use of TAM has been disputed for a while.   The authors should address the concerns of authors that published critiques   of TAM, and argue that their research model does not suffer from the   shortcomings mentioned there. See: Benbasat&Barki (2007): Quo vadis, TAM?   J. Assoc. Inf. Systems; and for health care: Sniehotta, Presseau &   Araujo-Soares (2014): Time to retire the theory of planned behaviour; Health   Psychology Review. 

Thank you for   your comments. We have discussed the importance of TAM in our introduction   section.

When introducing the constructs, please make   clear which constructs are compound constructs. The description of the   constructs should be improved, and the differences between them should be   pointed out. (e.g., 3.1 and 3.2 use partially similar wording)

The authors present six hypotheses; it is not   explicitly outlined in the paper which of these hypotheses are adopted and   which are rejected. Further, for each of these adoption/rejection decisions,   the consequences and reasons should be discussed. 

Thank you for   your comments. We have added a table to discuss the outcomes of the   hypotheses.

There are several paragraphs that are unclear   or imprecise due to language issues (long and strange sentences).

* line 32: IoT (not: IOT)

* line 33: wellbeing of our society: probably   you mean: wellbeing of citizens in our society.

* line 36: unclear; please explain why the   connected-self is relevant here.

* line 38: sentence is unclear. You say: the   devices have benefits for industries. Is this correct?

* line 42: remove word "elite"   (also non-elite athletes improve their performance using IoT devices)

* Line 45: incomplete sentence; why only   professionals? 

* in several occasions, the word   "arguably" is used. This does not always give meaning. For   instance, lines 60, 72, 79. Btw., the entire sentence starting in line 60 is   not comprehensible. 

* line 90: a small paragraph presenting the   contribution of the paper would be welcome. 

* line 110: what is "overlooking of   interpersonal influence" ?

* line 117: incomprehensible sentence

* line 126: incomprehensible sentence

* line 120: that their health is suffering:   please rephrase

In Table 1: Please check consistency (this   looks suspicious, but may be ok):

why are loading factors for both lines for   H.I.Accuracy the same? 

why are loading factors for both lines for   Privacy Protection the same? 

We have tried   our best to improve the presentation of our study. After subsequent checking,   the factor loadings of the mentioned constructs are really the same.

Round 2

Reviewer 2 Report

The authors have improved their paper. However, there are still some issues to be addressed.

The authors comment that they have addressed the comment about TAM being disputed. They have added a citation, but there is no reference to this citation. Although there are more discussions around TAM, I cannot find that this issue is further discussed.

The authors claim that they have introduced a table with the six hypotheses. I cannot find this table in the new version.

Please rephrase lines 267-269. "prior literatures" sounds strange.

In the new version, the authors mention that one hypothesis is not supportes without further comments on reasons. As this is a central finding in your research, you should at least mention that in your study privacy protection is not an antecedent that can explain adoption intention.

The conclusion (Section 8) is disjunct from the rest of the paper. In fact, nothing from the paper is used to support this conclusion. I would have expected that you present which factors can explain adoption behaviour, and what that means for the scientific community.

some grammar and spelling issues:

* line 102: provide, remove s

* line 109: health concerns, privacy concerns ...   use plural

* line 22: sentence is incomprehensible

* line 51: author name misspelled; please use umlaut u: Düking

Firstly, secondly, ... fourthly: inconsistent use. When counting to more than three, usually the forms without -ly are used. First, second, third, fourth, etc. can be used as adverbs and are preferred. There are many of these First, second, ... lists in your article; when reading, it becomes unclear what belongs to what.

see also line 76: here, you use the form without -ly, while the others in that list are with -ly...

line 79: it is quite unclear what no study has focused on ...

there are several sentences that need some attention, since there are grammar problems, or the sentences are too long to follow the flow in them. e.g., line 133: "applied to ..." is not comprehensible.

line 165: creating smart: two words

On page 9, have you checked the loading factors 0.947 and 0.952, as well as 0.942 on page 10 ? all of these appear twice each, which is rather unlikely with three decimals ?

note that there is a numbering issue with the manuscript. Numbering (both page and line numbering) starts over again after page 11.

second line 40: complicated sentence; incomprehensible.

second line 93: strange wording: ... are deemed to be useful ...

second lines 131-133: remove text from template.

Author Response

Response to reviewers’ comments

Dear Editor and reviewer,

We would like to thank you and the reviewers for reading our paper, for the considered comments provided and for giving us the opportunity to address those excellent comments.

Please find below our response to the comments from the reviewers. We have tried our best to seriously address all comments in our response and revised the manuscript accordingly. The result is certainly a much-improved paper.

The amendments to the revised manuscript are highlighted in BLUE in the manuscript.

We look forward to the positive outcome of this revision of the manuscript.

Kind regards,

The authors

Reviewer 2’s Comments

Response to Reviewer 2

The authors have improved their paper. However, there are still some   issues to be addressed.

The authors comment that they have addressed the comment about TAM   being disputed. They have added a citation, but there is no reference to this   citation. Although there are more discussions around TAM, I cannot find that   this issue is further discussed.

The authors claim that they have introduced a table with the six   hypotheses. I cannot find this table in the new version.

Please rephrase lines 267-269. "prior literatures" sounds   strange.

In the new version, the authors mention that one hypothesis is not   supportes without further comments on reasons. As this is a central finding   in your research, you should at least mention that in your study privacy   protection is not an antecedent that can explain adoption intention.

The conclusion (Section 8) is disjunct from the rest of the paper. In   fact, nothing from the paper is used to support this conclusion. I would have   expected that you present which factors can explain adoption behaviour, and   what that means for the scientific community.

some grammar and spelling issues:

* line 102: provide, remove s

* line 109: health concerns, privacy concerns ...   use plural

* line 22: sentence is incomprehensible

* line 51: author name misspelled; please use umlaut u: Düking

Firstly, secondly, ... fourthly: inconsistent use. When counting to   more than three, usually the forms without -ly are used. First, second,   third, fourth, etc. can be used as adverbs and are preferred. There are many   of these First, second, ... lists in your article; when reading, it becomes   unclear what belongs to what.

see also line 76: here, you use the form without -ly, while the others   in that list are with -ly...

line 79: it is quite unclear what no study has focused on ...

there are several sentences that need some attention, since there are   grammar problems, or the sentences are too long to follow the flow in them.   e.g., line 133: "applied to ..." is not comprehensible.

line 165: creating smart: two words

On page 9, have you checked the loading factors 0.947 and 0.952, as   well as 0.942 on page 10 ? all of these appear twice each, which is rather   unlikely with three decimals ?

note that there is a numbering issue with the manuscript. Numbering   (both page and line numbering) starts over again after page 11.

second line 40: complicated sentence; incomprehensible.

second line 93: strange wording: ... are deemed to be useful ...

second lines 131-133: remove text from template.

Thank you for your comments.  

Regarding the table with the   six hypotheses, please refer to table 3, page 11.

Regarding the critique of   TAM, we have deleted the critique. 

The paragraph (Lines 267 –   269) was rephrased. “Prior literatures” was changed to “Previous studies”.

We have added a paragraph in   section 6.1, highlighted in blue. Surprisingly, we found that the impact of privacy protection on perceived   usefulness is non-significant, suggesting that privacy protection is not an   antecedent of perceived usefulness of healthcare wearable technology.

We have deleted the   conclusion, following your comments.

Regarding the grammatical   issues, we have amended the writing (highlighted in blue). 

The numbers in table 1 were   changed, highlighted in blue.

The managing editor /   editorial assistants will edit the numbering of lines.
